# RNA-guided retargeting of *Sleeping Beauty* transposition in human cells

**Adrian Kovač[1], Csaba Miskey[1], Michael Menzel[2], Esther Grueso[1], Andreas Gogol-Döring[2], Zoltán Ivics[1]\***

[1]Transposition and Genome Engineering, Division of Medical Biotechnology, Paul Ehrlich Institute, Langen, Germany; [2]University of Applied Sciences, Giessen, Germany

**Abstract** An ideal tool for gene therapy would enable efficient gene integration at predetermined sites in the human genome. Here we demonstrate biased genome-wide integration of the *Sleeping Beauty* (SB) transposon by combining it with components of the CRISPR/Cas9 system. We provide proof-of-concept that it is possible to influence the target site selection of SB by fusing it to a catalytically inactive Cas9 (dCas9) and by providing a single guide RNA (sgRNA) against the human *Alu* retrotransposon. Enrichment of transposon integrations was dependent on the sgRNA, and occurred in an asymmetric pattern with a bias towards sites in a relatively narrow, 300 bp window downstream of the sgRNA targets. Our data indicate that the targeting mechanism specified by CRISPR/Cas9 forces integration into genomic regions that are otherwise poor targets for SB transposition. Future modifications of this technology may allow the development of methods for specific gene insertion for precision genetic engineering.

## Introduction

The ability to add, remove or modify genes enables researchers to investigate genotype-phenotype relationships in biomedical model systems (functional genomics), to exploit genetic engineering in species of agricultural and industrial interest (biotechnology) and to replace malfunctioning genes or to add functional gene sequences to cells in order to correct diseases at the genetic level (gene therapy).

One option for the insertion of genetic cargo into genomes is the use of integrating vectors. The most widely used integrating genetic vectors were derived from retroviruses, in particular from γ-retroviruses and lentiviruses (*Escors and Breckpot, 2010*). These viruses have the capability of shuttling a transgene into target cells and stably integrating it into the genome, resulting in long-lasting expression. Transposons represent another category of integrating vector. In contrast to retroviruses, transposon-based vectors only consist of a transgene flanked by inverted terminal repeats (ITRs) and a transposase enzyme, the functional equivalent of the retroviral integrase (*Tipanee et al., 2017*). For DNA transposons, the transposase enzymes excise genetic information flanked by the ITRs from the genome or a plasmid and reintegrate it at another position (*Figure 1A*). Thus, transposons can be developed as non-viral gene delivery tools (*Ivics et al., 2009*) that are simpler and cheaper to produce, handle and store than retroviral vectors (*Hudecek and Ivics, 2018*). The absence of viral proteins may also prevent immune reactions that are observed with adeno-associated virus (AAV)-based vectors (*Mingozzi and High, 2011*; *Hareendran et al., 2013*). The *Sleeping Beauty* (SB) transposon is a Class II DNA transposon, whose utility has been demonstrated in pre-clinical (reviewed in *Tipanee et al., 2017*; *Hudecek et al., 2017*) as well as clinical studies (*Singh et al., 2008*; *Kebriaei et al., 2016*; reviewed in *Narayanavari and Izsvák, 2017*). It is active across a wide range of cell types (*Ivics et al., 1997*; *Izsvák et al., 2000*) and hyperactive variants

\*For correspondence:
zoltan.ivics@pei.de

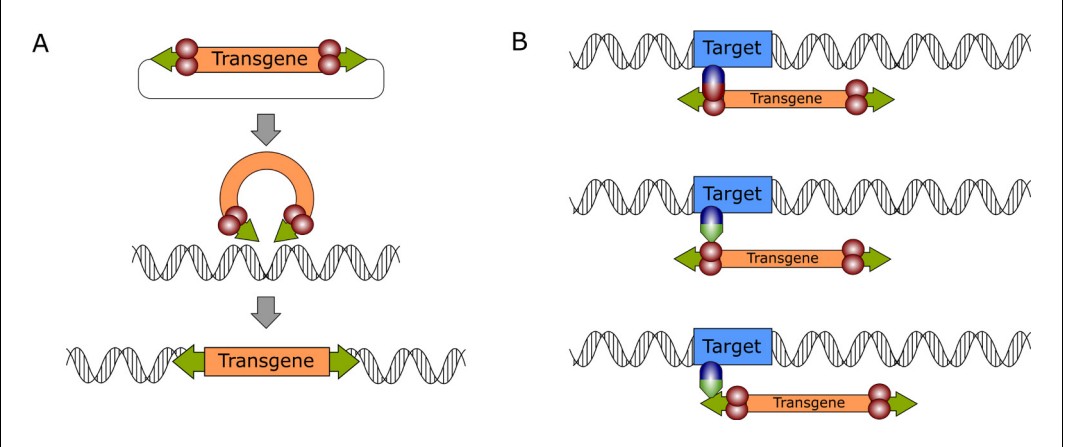

**Figure 1.** General mechanism of DNA transposition and molecular strategies for targeted gene integration. (**A**) The transpositional mechanism of a DNA transposon in a biotechnological context. The transgene, which is flanked by transposon ITRs (green arrows) is excised from a plasmid by the transposase enzyme (red spheres), which is supplied in trans. The genetic cargo is then integrated in the target genome. (**B**) Transposition can be retargeted by foreign factors that can be DNA-binding domains (blue spheres) directly fused to the transposase or to adapter domains (green triangles) that interact either with the transposase (middle) or the transposon DNA (bottom).

such as the SB100X transposase catalyze gene transfer in human cells with high efficiency (*Mátés et al., 2009*).

The main drawback of integrating vectors is their unspecific or semi-random integration (*Kovač and Ivics, 2017*). For example, lentiviral or γ-retroviral vectors actively target genes or transcriptional start sites (*Schröder et al., 2002*; *Cohn et al., 2015*; *Wu et al., 2003*; *Cattoglio et al., 2010*; *Mitchell et al., 2004*). In contrast, the SB transposon displays a great deal of specificity of insertion at the primary DNA sequence level – almost exclusively integrating into TA dinucleotides (*Vigdal et al., 2002*) – but inserts randomly on a genome-wide scale (*Yant et al., 2005*; *Moldt et al., 2011*; *Huang et al., 2010*; *Zhang et al., 2013*). Thus, because all of these vectors can potentially integrate their genetic cargo at a vast number of sites in the genome, the interactions between the transgene and the target genome are difficult to predict. For example, the position of a transgene in the genome can have an effect on the expression of the transgene, endogenous genes or both (*Bestor, 2000*; *Ellis, 2005*; *Hacein-Bey-Abina et al., 2003*; *Stein et al., 2010*; *Howe et al., 2008*; *Cavazzana-Calvo et al., 2010*). Especially in therapeutic applications, controlled transgene expression levels are important, because low expression levels could fail to produce the desired therapeutic effect, while overexpression might have deleterious effects on the target cell. Perhaps more dramatic are the effects transgenes might have on the genome. Insertion of transgenes can disrupt genomic regulation, either by direct insertional mutagenesis of cellular genes or regulatory elements, or by upregulation of genes in the vicinity of the integration site. In the worst case, this can result in overexpression of a proto-oncogene or disruption of a tumor suppressor gene; both of these outcomes can lead to transformation of the cell and tumor formation in the patient.

An alternative technology used in genetic engineering is based on targeted nucleases; the most commonly used nuclease families are zinc finger nucleases (ZFNs) (*Urnov et al., 2010*), transcription activator-like effector-based nucleases (TALENs) (*Ousterout and Gersbach, 2016*) and the CRISPR/Cas system (*Doudna and Charpentier, 2014*). All of these enzymes perform two functions: they have a DNA-binding domain (DBD) that recognizes a specific target sequence and a nuclease domain that cleaves the target DNA once it is bound. While for ZFNs and TALENs target specificity is determined by their amino acid sequence, Cas nucleases need to be supplied with a single guide RNA (sgRNA) that determines their target specificity (*Jinek et al., 2012*). This makes the CRISPR/Cas system significantly more flexible than other designer nucleases.

A double-strand break (DSB) in a target cell is usually repaired by the cell's DNA repair machinery, either via non-homologous end-joining (NHEJ) or homologous recombination (HR) (*Mao et al., 2008*; *Kakarougkas and Jeggo, 2014*). The NHEJ pathway directly fuses the two DNA ends together. Due to the error-prone nature of this reaction, short insertions or deletions (indels) are often produced. Because this in turn often results in a frame-shift in a coding sequence, this process can be used to effectively knock out genes in target cells. If a DNA template is provided along with the nuclease, a DSB can also be repaired by the HR pathway. This copies the sequence information from the repair template into the target genome, allowing replacement of endogenous sequences or knock-in of completely new genes (*Porteus and Baltimore, 2003*). Thus, knock-in of exogenous sequences into a genetic locus is a cumulative outcome of DNA cleavage by the nuclease and HR by the cell. However, the efficiency of the HR pathway is low compared to the efficiency of the nuclease (*Lieber, 2010*). This bottleneck means that targeted nucleases are highly efficient at knocking out genes (*Hockemeyer et al., 2009*; *Hockemeyer et al., 2011*), but less efficient at inserting DNA (*Aird et al., 2018*), particularly when compared to the integrating viral and non-viral vectors mentioned previously. Thus, integrating vectors and nuclease-based approaches to genome engineering have overlapping but distinct advantages and applications: nuclease-based approaches are site-specific and efficient at generating knock-outs, while integrating vectors are unspecific but highly efficient at generating knock-ins.

Based on the features outlined above, it is plausible that the specific advantages of both approaches (designer nucleases and integrating vector systems) could be combined into a single system with the goal of constructing a gene delivery tool which inserts genetic material into the target cell's genome with great efficiency and at the same time in a site-specific manner. Indeed, by using DBDs to tether integrating enzymes (retroviral integrases or transposases) to the desired target, one can combine the efficient, DSB-free insertion of genetic cargo with the target specificity of designer nucleases (reviewed in *Kovač and Ivics, 2017*). In general, two approaches can be used to direct transposon integrations by using a DBD: direct fusions or adapter proteins (*Kovač and Ivics, 2017*). In the direct fusion approach, a fusion protein of a DBD and the transposase is generated to tether the transposase to the target site (*Figure 1B*, top). However, the overall transposase activity of these fusion proteins is often reduced. Alternatively, an adapter protein can be generated by fusing the DBD to a protein domain interacting with the transposase or the transposon (*Figure 1B*, middle and bottom, respectively). Several transposon systems, notably the SB and the *piggyBac* systems have been successfully targeted to a range of exogenous or endogenous loci in the human genome (*Voigt et al., 2012*; *Ammar et al., 2012*; *Ivics et al., 2007*; reviewed in *Kovač and Ivics, 2017*). However, a consistent finding across all targeted transposition studies is that while some bias can be introduced to the vector's integration profile, the number of targeted integrations is relatively low when compared to the number of untargeted background integrations (*Kovač and Ivics, 2017*).

In the studies mentioned above, targeting was achieved with DBDs including ZFs or TALEs, which target a specific sequence determined by their structure. However, for knock-outs, the CRISPR/Cas system is currently the most widely used technology due to its flexibility in design. A catalytically inactive variant of Cas9 called dCas9 ('dead Cas9', containing the mutations D10A and H840A), has previously been used to target enzymes including transcriptional activators (*Konermann et al., 2015*; *Maeder et al., 2013*; *Perez-Pinera et al., 2013*), repressors (*Yeo et al., 2018*; *Gilbert et al., 2013*), base editors (*Eid et al., 2018*; *Gehrke et al., 2018*) and others (*Chaikind et al., 2016*; *Halperin et al., 2018*) to specific target sequences. Using dCas9 as a targeting domain for a transposon could combine this great flexibility with the advantages of integrating vectors. By using the *Hsmar1* human transposon (*Miskey et al., 2007*), a 15-fold enrichment of transposon insertions into a 600 bp target region was observed in an in vitro plasmid-to-plasmid assay employing a dCas9-*Hsmar1* fusion (*Bhatt and Chalmers, 2019*). However, no targeted transposition was detected with this system in bacterial cells. A previous study failed to target the *piggyBac* transposon into the *HPRT* gene with CRISPR/Cas9 components in human cells, even though some targeting was observed with other DBDs (*Luo et al., 2017*). However, in a recent study, some integrations were successfully biased to the *CCR5* locus using a dCas9-*piggyBac* fusion (*Hew et al., 2019*). Two additional recent studies showed highly specific targeting of bacterial Tn*7*-like transposons by an RNA-guided mechanism, but only in bacterial cells (*Strecker et al., 2019*; *Klompe et al., 2019*).

Previous studies have established that foreign DBDs specifying binding to both single-copy as well as repetitive targets can introduce a bias into SB's insertion profile, both as direct fusions with

the transposase and as fusions to the N57 targeting domain. N57 is an N-terminal fragment of the SB transposase encompassing the N-terminal helix-turn-helix domain of the SB transposase with dual DNA-binding and protein dimerization functions (*Izsvák et al., 2002*). Fusions of N57 with the tetracycline repressor (TetR), the E2C zinc finger domain (*Beerli et al., 1998*), the ZF-B zinc finger domain and the DBD of the Rep protein of AAV were previously shown to direct transposition catalyzed by wild-type SB transposase to genomically located tetracycline operator (TetO) sequences, the *erbB-2* gene, endogenous human L1 retrotransposons and Rep-recognition sequences, respectively (*Voigt et al., 2012*; *Ammar et al., 2012*; *Ivics et al., 2007*). Here, we present proof-of-principle evidence that integrations of the SB transposon system can be biased towards endogenous *Alu* retrotransposons using dCas9 as a targeting domain in an sgRNA-dependent manner.

## Results

### Design and validation of sgRNAs targeting single-copy and repetitive sites in the human genome

Two different targets were chosen for targeting experiments: the *HPRT* gene on the X chromosome and *Alu*Y, an abundant (~130000 elements per human genome) and highly conserved family of *Alu* retrotransposons (*Bennett et al., 2008*). Four sgRNAs were designed to target the *HPRT* gene (*Figure 2A*), one of them (sgHPRT-0) binding in exon 7 and three (sgHPRT-1 – sgHPRT-3) in exon 3. Three sgRNAs were designed against *Alu*Y (*Figure 2D*), the first two (sgAluY-1 and sgAluY-2) binding at or near the conserved A-box of the Pol III promoter that drives *Alu* transcription and the third (sgHPRT-3) against the A-rich stretch that separates the two monomers in the full-length *Alu* element.

The *HPRT*-specific sgRNAs were tested by transfecting human HCT116 cells with a Cas9 expression plasmid and expression plasmids that supply the different *HPRT*-directed sgRNAs. Disruption of the *HPRT* coding sequence by NHEJ was measured by selection with 6-TG, which is lethal to cells in which the *HPRT* gene is intact. Thus, the number of 6-TG-resistant cell colonies obtained in each sample is directly proportional to the extent to which the *HPRT* coding sequence is mutagenized and functionally inactivated. Two sgRNAs (sgHPRT-0, sgHPRT-1) resulted in strong, significant increases in disruption levels (p≤0.001), while sgHPRT-2 failed to increase disruption over the background level, and sgHPRT-3 induced weak but significant disruption (p≤0.05). (*Figure 2B*). The efficiency of sgHPRT-0 was further tested with a TIDE assay, which compares sequence data from two standard capillary (Sanger) sequencing reactions, thereby quantifying editing efficacy in terms of indels in the targeted DNA in a cell pool. As measured by TIDE, sgHPRT-0 yielded a total editing efficiency of 57.1% (*Figure 2C*).

The activities of the *Alu*Y-directed sgRNAs were first analyzed by an in vitro cleavage assay. Incubation of human genomic DNA (gDNA) with purified Cas9 protein and in vitro transcribed sgRNAs showed detectable fragmentation of gDNA for sgAluY-1 and sgAluY-2 (*Figure 2E*). gDNA digested with Cas9 and sgAluY-1 was purified, cloned into a plasmid vector and the sequences of the plasmid-genomic DNA junctions were determined. Twelve of 32 sequenced genomic junctions could be mapped to the *Alu*Y sequence upstream of the cleavage site and 19 could be mapped to the sequence immediately downstream (as defined by the direction of *Alu* transcription). A consensus sequence generated by aligning the 12 or 19 sequences showed significant similarity to the *Alu*Y consensus sequence (*Figure 2F*), demonstrating that the DNA fragmentation was indeed the result of Cas9-mediated cleavage. The sequence composition also revealed that mismatches within the sgRNA binding sequence are tolerated to some extent, while the conserved GG dinucleotide of the NGG PAM motif did not show any sequence variation (*Figure 2F*). In sum, the data establish functional sgRNAs against the single-copy *HPRT* locus (by sgHPRT-0) and against the repetitive *Alu*Y sequence (by sgAluY-1).

### Generation of Cas9 fusion constructs and their functional validation

Three different targeting constructs were generated to test both the direct fusion and the adapter protein approaches described above. For the direct fusion, the entire coding sequence of SB100X, a hyperactive version of the SB transposase (*Mátés et al., 2009*), was inserted at the C-terminus of the dCas9 sequence (*Figure 3A*, top). We only made an N-terminal SB fusion, because C-terminal

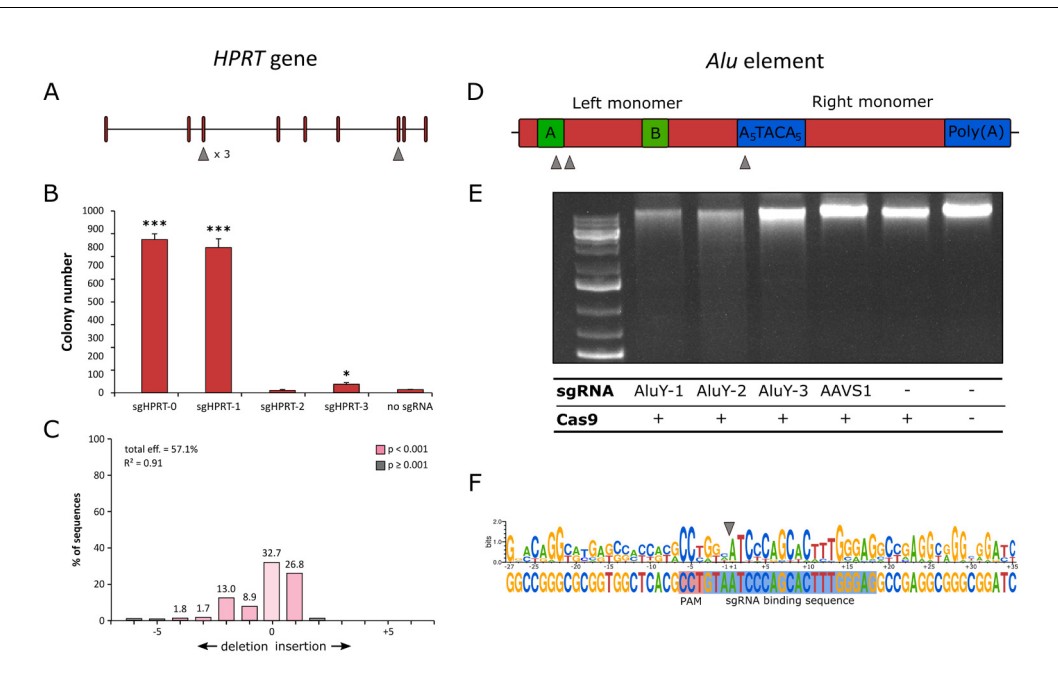

**Figure 2.** CRISPR/Cas9 components and their validation for transposon targeting. (**A**) Schematic exon-intron structure of the *HPRT* gene and positions of the sgRNA binding sites. (**B**) Numbers of 6-TG resistant colonies after treatment with Cas9 and *HPRT*-directed sgRNAs. Significance is calculated in comparison to the no sgRNA sample (n = 3, biological replicates for all samples, *p≤0.05, ***p≤0.001, error bars represent SEM). (**C**) Indel spectrum of the *HPRT* locus after treatment with Cas9 and sgHPRT-0, as determined by TIDE assay. (**D**) Structure of an *Alu* element and relative positions of sgRNA binding sites. (**E**) Agarose gel electrophoresis of human gDNA digested with Cas9 and *AluY*-directed sgRNAs. An sgRNA targeting the human *AAVS1* locus (a single-copy target) as well as samples containing no Cas9 or no sgRNA were included as negative controls. (**F**) Sequence logo generated by aligning sequenced gDNA ends after fragmentation with Cas9 and sgAluY-1 (the sequence represents the top strand targeted by the sgRNA). The position of the sgRNA-binding site and PAM is indicated by blue and red background, respectively. The cleavage site is marked by the gray arrow. The sequence upstream of the cleavage site is generated from 12 individual sequences, the sequence downstream is generated from 19 individual sequences. The bottom sequence represents the *AluY* consensus sequence.

tagging of the transposase enzyme completely abolishes its activity (*Yant et al., 2007*; *Ivics et al., 2007*; *Wilson et al., 2005*). For adapter proteins, the N57 domain was inserted at the C-terminus as well as at the N-terminus of dCas9 (*Figure 3A*, middle and bottom, respectively). N57 interacts both with SB transposase molecules and the SB transposon ITRs, and could thus potentially use multiple mechanisms for targeting, as outlined in *Figure 1B*. A flexible linker KLGGGAPAVGGGPK (*Szuts and Bienz, 2000*) that was previously validated in the context of SB transposase fusions to ZFs (*Voigt et al., 2012*) and to Rep (*Ammar et al., 2012*) DBDs was introduced between dCas9 and the full-length SB100X transposase or the N57 targeting domain (*Figure 3A*). All three protein fusions were cloned into all-in-one expression plasmids that allow co-expression of the dCas9-based targeting factors with sgRNAs.

Western blots using an antibody against the SB transposase verified the integrity and the expression of the fusion proteins (*Figure 3B*). In order to verify that the dCas9-SB100X direct fusion retained sufficient transpositional activity, we measured its efficiency at integrating a puromycin-marked transposon into HeLa cells, and compared its activity to the unfused SB100X transposase (*Figure 4A*). We found that the fusion construct dCas9-SB100X was approximately 30% as active as unfused SB100X. To verify that N57 retains its DNA-binding activity in the context of the dCas9 fusions, we performed an EMSA experiment using a short double-stranded oligonucleotide corresponding to the N57 binding sequence in the SB transposon (*Figure 4B*). Binding could be detected for the dCas9-N57 fusion, but not for N57-dCas9. For this reason, the N57-dCas9 construct was excluded from the subsequent experiments. The DNA-binding ability of the dCas9 domain in the

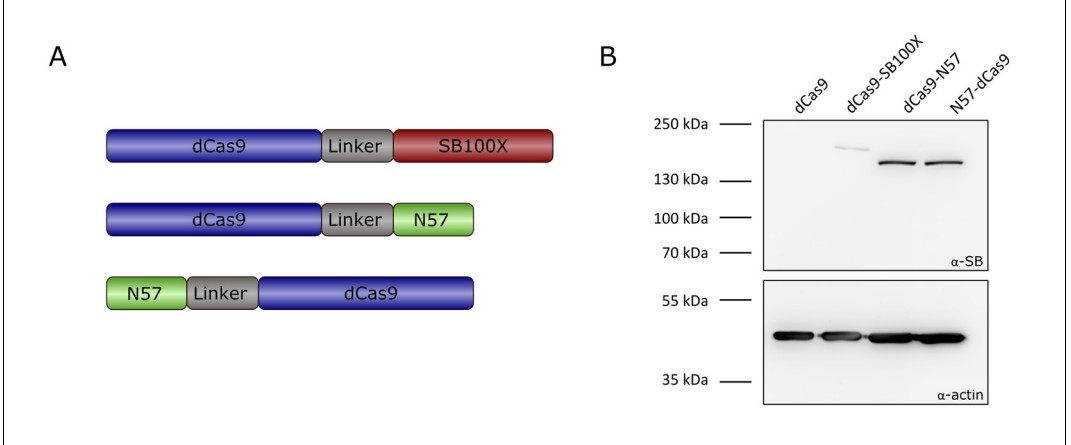

**Figure 3.** Transposase-derived targeting factors. (**A**) Schematic representation of the targeting constructs. (**B**) Western blot of proteins expressed by the targeting constructs. The top half of the membrane was treated with α-SB antibody, the bottom half was treated with α-actin as a loading control. dCas9 was included as a negative control, and is therefore not expected to produce a signal with an antibody against the SB transposase. Expected sizes were 202.5 kDa for dCas9-SB100X and 169.7 kDa for dCas9-N57 and N57-dCas9.

fusion constructs was not tested directly. Instead, analogous constructs containing catalytically active Cas9 were generated and tested for cleavage activity. The activities of these fusion constructs were determined by measuring the disruption frequency of the *HPRT* gene by selection with 6-TG, as described above. The cleavage efficiencies of both Cas9-SB100X and Cas9-N57 were ~30% of unfused Cas9 in the presence of sgHPRT-0 (*Figure 4C*). Because binding of the Cas9 domain to its target DNA is a prerequisite for DNA cleavage, we infer that cleavage-competent fusion proteins are also able to bind to target DNA. Collectively, these data establish that our dCas9 fusion proteins i) are active in binding to the target DNA in the presence of sgRNA; ii) they retain transposition activity (for the fusion with the full-length SB100X transposase); and iii) they can bind to the transposon DNA (for the fusion with the C-terminal N57 targeting domain), which constitute the minimal requirements for targeted transposition in the human genome.

## RNA-guided *Sleeping Beauty* transposition in the human genome

Having established functionality of our multi-component transposon targeting system, we next analyzed the genome-wide patterns of transposon integrations catalyzed by the different constructs. Transposition reactions were performed in human HeLa cells with dCas9-SB100X or dCas9-N57 + SB100X complemented with sgRNAs (sgHPRT-0 or sgAluY-1) (*Figure 5*). As a reference dataset, we generated independent insertions in the presence of sgL1-1 that targets the 3'-terminus of human L1 retrotransposons (*Figure 5—figure supplement 1*). This sgRNA was validated for in vitro cleavage by Cas9, and was found to yield some enrichment of SB insertions within a 500 bp window downstream of the sgRNA binding sites (*Figure 5—figure supplement 1*), although without the power of statistical significance. The sgL1-1 insertion site dataset was nevertheless useful to serve as a negative control obtained with an unrelated sgRNA. Integration libraries consisting of PCR-amplified transposon-genome junctions were generated and subjected to high-throughput sequencing. Recovered reads were aligned to the human genome (hg38 assembly) to generate lists of insertion sites. In order to quantify the targeting effects, we defined targeting windows of different sizes around the sgRNA binding sites (*Figure 5A*). The fraction of overall insertions into each targeting window was calculated (*Figure 5B*), and these ratios were compared to those obtained with the negative control (same targeting construct with sgL1-1) (*Figure 5C and D*). For the *HPRT* locus, no insertion was recovered within 5 kb in either direction from the sgHPRT-0 binding site in our dataset (data not shown). We conclude that either targeting of this single-copy locus was not possible with the current system, or that the number of insertion sites recovered (<1000 insertions) was too low to provide the necessary resolution for detecting an effect.

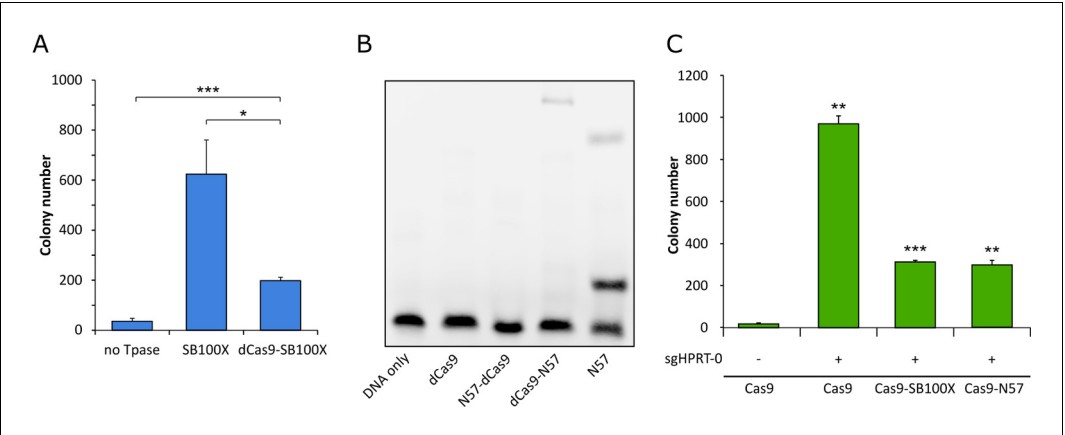

**Figure 4.** Functional testing of dCas9 fusions. (**A**) Numbers of puromycin-resistant colonies in the transposition assay. The dCas9-SB100X fusion protein catalyzes ~30% as many integration events as unfused SB100X transposase (n = 3, biological replicates, *p≤0.05, ***p≤0.001, error bars represent SEM). (**B**) EMSA with dCas9-N57 fusion proteins. dCas9 serves as negative control, N57 as positive control. Binding can be detected for dCas9-N57, but not for N57-dCas9. The upper band in the positive control lane is likely a multimeric complex of DNA-bound N57 molecules, in line with N57's documented activity in mediating protein-protein interaction between transposase subunits and in forming higher-order complexes (*Izsvák et al., 2002*). (**C**) Numbers of 6-TG resistant colonies after Cas9 cleavage. No disruption of the *HPRT* gene, as measured by 6-TG resistance over background, can be detected without the addition of an sgRNA. In the presence of sgHPRT-0, all Cas9 constructs cause significant disruption of the *HPRT* gene (n = 3, biological replicates, **p≤0.01, ***p≤0.001, error bars represent SEM).

Next, integration site datasets generated with dCas9-N57 + SB100X + sgAluY-1 (*Figure 5— source data 1*, 13269 insertions), dCas9-N57 + SB100X + sgL1-1 (*Figure 5—source data 2*, 12350 insertions) as well as dCas9-SB100X and sgAluY-1 (*Figure 5—source data 3*, 1463 insertions) and dCas9-SB100X and sgL1-1 (*Figure 5—source data 4*, 2769 insertions) were compared (*Figure 5B*). The sgAluY-1 sgRNA has a total of 299339 target sites in the human genome (hg38) (the number of sites exceeds the number of *Alu*Y elements due to high conservation, and therefore presence in other *Alu* subfamilies). We found some enrichment (ca. 15%) for dCas9-N57 + SB100X in a window of 200 bp around the target sites and dCas9-SB100X insertions are slightly enriched in a window of 500 bp (ca. 20%) (*Figure 5C*), although neither change was statistically significant. To further investigate the distribution of insertions around the target sites, we decreased the size of the targeting windows and counted insertions in up- and downstream windows independently. We only found a modest enrichment with dCas9-N57, and the pattern seemed to be relatively symmetrical in a window from −150 bp to +150 bp with respect to the sgRNA binding sites (*Figure 5D*). However, with dCas9-SB100X, we found that the enrichment occurred almost exclusively downstream of the target sites, within the *Alu*Y element. We detected statistically significant enrichment in the insertion frequencies in a window spanning a 300 bp region downstream of the sgRNA target sites (~1.5 fold enrichment, p=0.019) (*Figure 5D*). We also detected enrichment near target loci similar to the target site (with one mismatch), although not statistically significant (*Figure 5E*). This result is in agreement with the finding that the specificity of dCas9 binding is lower than that of Cas9 cleavage (*Jiang and Doudna, 2017*).

Intriguingly, plotting the overall insertion frequencies around the target sites revealed that the SB insertion machinery generally disfavors loci downstream of the sgAluY-1 binding sequences (*Figure 6A*). These results together with the asymmetric pattern of integrations next to the target sites prompted us to investigate properties of the genomic loci around the sgRNA target sites. Along this line, we next set out to investigate the target nucleotides of the transposons in the targeted segments. To our surprise, we found that the TA dinucleotide frequency in the targeted region is in fact lower than in the neighboring segments (*Figure 6B*). Along these findings, comparison of the nucleotide composition of the targeted *vs* non-targeted insertion sites revealed that the integrations within the *Alu* sequences are enforced to take place at TA sequences that only weakly

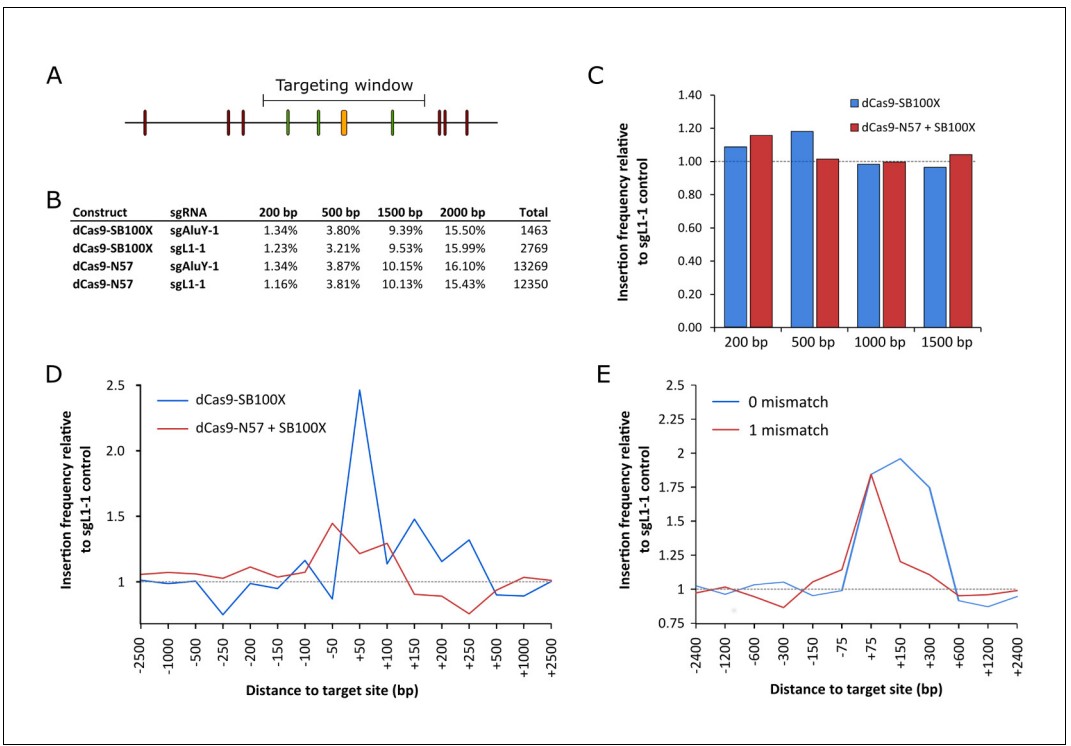

**Figure 5.** RNA-guided *Sleeping Beauty* transposition in human cells. (**A**) Schematic representation of the analysis of SB retargeting. Targeting windows are defined as DNA extending a certain number of base pairs upstream or downstream of the sgRNA target sites (yellow – sgRNA target, green – 'hit' insertion, red – 'miss' insertion). (**B**) Percentages of integrations recovered from windows of different sizes along with the total numbers of integrations in the respective libraries. (**C**) Insertion frequencies relative to a dataset obtained with sgL1-1, in windows of various sizes around the targeted sites. Slight enrichment can be observed in a 200 bp window with dCas9-N57 and in a 500 bp window with dCas9-SB100X, although neither enrichment is statistically significant. The windows are cumulative, that is the 500 bp window also includes insertions from the 200 bp window. (**D**) Insertion frequencies in windows of various sizes, relative to a dataset obtained with sgL1-1, upstream and downstream of the target sites. Enrichment with dCas9-SB100X occurs downstream of the sgRNA target site, within a total insertion window of 300 bp (~1.5 fold enrichment, p=0.019). (**E**) The effect of the number of mismatches on the targeting efficiency of dCas9-SB100X. Relative insertion frequencies of the dCas9-SB100X sample into cumulative windows around perfectly matched target sites as well as sites with a single mismatch.

The online version of this article includes the following source data and figure supplement(s) for figure 5:

**Source data 1.** *Sleeping Beauty* transposon integration sites obtained with dCas9-N57+SB100X and sgAluY-1.
**Source data 2.** *Sleeping Beauty* transposon integration sites obtained with dCas9-N57+SB100X and sgL1-1.
**Source data 3.** *Sleeping Beauty* transposon integration sites obtained with dCas9-SB100X and sgAluY-1.
**Source data 4.** *Sleeping Beauty* transposon integration sites obtained with dCas9-SB100X and sgL1-1.
**Figure supplement 1.** Design, in vitro validation and impact of sgRNAs against human L1 retrotransposon sequences.

match the preferred ATATATAT consensus palindrome (*Figure 6—figure supplement 1*). Thus, targeting occurs into DNA that is per se disfavored by the SB transposition machinery. Since the nucleotide composition of the targeted regions is remarkably different from that of the neighboring sequences and given that nucleosome positioning in the genome is primarily driven by sequence (*Segal et al., 2006*), we next investigated nucleosome occupancy of the target DNA. Nucleosome occupancy was predicted in 2 kb windows on 20000 random target sequences and on all the insertion sites of the non-targeted condition (unfused SB100X). This analysis recapitulated our previous finding showing that SB disfavors integrating into nucleosomal DNA (*Gogol-Döring et al., 2016*). Additionally, in agreement with previous findings of others (*Englander and Howard, 1995*; *Tanaka et al., 2010*), we found that these *Alu*Y sequences are conserved regions for nucleosome formation (*Figure 6C*). These results can explain the overall drop in insertion frequency of SB into

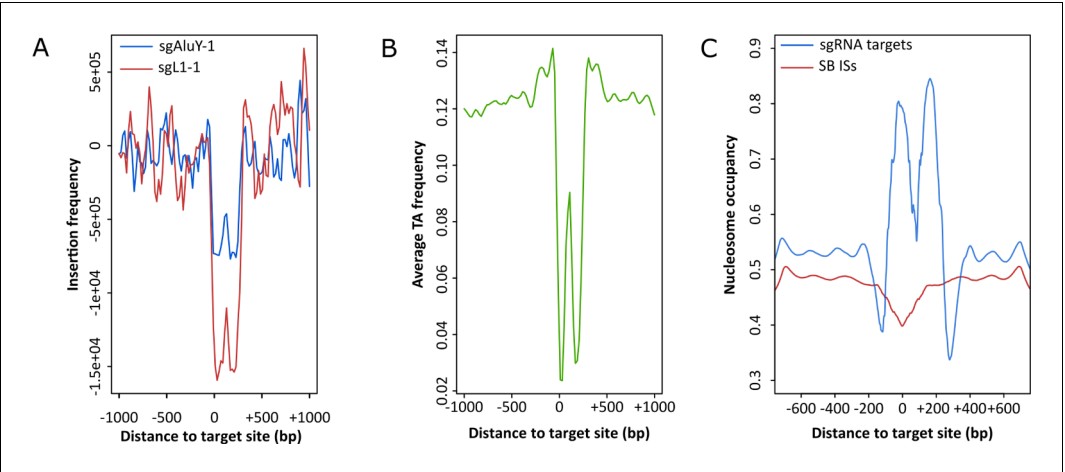

**Figure 6.** Analysis of targeted chromosomal regions. (**A**) Insertion frequencies of the targeted (blue) and non-targeted (red) dataset show that statistically significant (p=0.019) enrichment occurs within a 300 bp window downstream of sites targeted by sgAluY-1, which is generally disfavored for SB integration. (**B**) Reduced average TA di-nucleotide frequency the targeted 300 bp window. (**C**) Computationally predicted nucleosome occupancy around the sites targeted by sgAluY-1 (blue) and around untargeted SB insertion sites (ISs, red).
The online version of this article includes the following figure supplement(s) for figure 6:

**Figure supplement 1.** Sequence logos generated from sequences around insertion sites catalyzed by dCas9-SB100X with sgAluY-1 within the 300 bp targeting window (left) and outside of the window (right). The left logo has higher variation at most positions, because of the lower number of insertions.

these regions. In sum, the data above establish weak, sgRNA-dependent enrichment of SB transposon integrations around multicopy genomic target sites in the human genome.

## Discussion

We demonstrate in this study that the insertion pattern of the SB transposase can be influenced by fusion to dCas9 as an RNA-guided targeting domain in human cells, and as a result be weakly biased towards sites specified by an sgRNA that targets a sequence in the *Alu*Y repetitive element. We consider it likely that the observed enrichment of insertions next to sgRNA-targeted sites is an underestimate of the true efficiency of transposon targeting in our experiments, because our PCR procedure followed by next generation sequencing and bioinformatic analysis cannot detect independent targeting events that had occurred at the same TA dinucleotide in the human genome. While enrichment observed with dCas9-N57 was very weak and not statistically significant, the enrichment by dCas9-SB100X was more pronounced, and occurred in a relatively narrow window in the vicinity of the sites specified by the sgRNA. This observation is consistent with physical docking of the transpositional complex at the targeted sites, and suggests that binding of dCas9 to its target sequence and integration by the SB transposase occur within a short timeframe. We further detect an asymmetric distribution of insertions around the target sites. Asymmetric distributions of targeted insertions have been previously found in a study using the ISY100 transposon (which, like SB, is a member of the Tc1/*mariner* transposon superfamily) in combination with the ZF domain Zif268 in *E. coli* (***Feng et al., 2010***) and in experiments with dCas9-*Hsmar1* fusions in vitro (***Bhatt and Chalmers, 2019***). Enrichment mainly occurring downstream of the sgRNA target site in our experiments was somewhat surprising, as domains fused to the C-terminus of Cas9 are expected to be localized closer to the 5'-end of the target strand (***Oakes et al., 2014***), or upstream of the sgRNA binding site. The fact that SB100X is connected with dCas9 by a relatively long, flexible linker could explain why enrichment can occur on the other side of the sgRNA binding site, but it does not explain why enrichment on the 'far side' seems to be more efficient. Against expectations, we found that the window in which the highest enrichment occurs represents a disfavored target for SB transposition (***Figure 6A***), likely because it is TA-poor (***Figure 6B***) – the *Alu*Y consensus sequence has a GC

content of 63% (*Price et al., 2004*) – and nucleosomal (*Figure 6C*). It is possible that the targeting effect in this window is more pronounced than on the other side of the sgRNA target site because there are fewer background insertions obscuring a targeting effect.

Unlike in our earlier studies establishing biased transposon integration by the N57 targeting peptide fused to various DBDs (*Ivics et al., 2007*; *Voigt et al., 2012*; *Ammar et al., 2012*), our dCas9-N57 fusion apparently did only exert a minimal effect on the genome-wide distribution of SB transposon insertions (*Figure 5*). Because Cas9-N57 is active in cleavage (*Figure 4C*) and dCas9-N57 is active in binding to transposon DNA (*Figure 4B*), this result was somewhat unexpected. We speculate that addition of a large protein (dCas9 is 158 kDa) to the N-terminus of a relatively small polypeptide of 57 amino acids masks its function to some extent. Indeed, TetR, the ZF-B protein and Rep DBD that were used previously with success in conjunction with N57 are all far smaller than dCas9. The binding activity of dCas9-N57 to transposon DNA, though detectable by EMSA, may have been too weak to effectively recruit the components of the SB system to the target site.

Our data reveal some of the important areas where refined molecular strategies as well as reagents may yield higher targeting efficiencies. First, the difficulty of targeting to a single location, in this case the *HPRT* gene, might be associated with characteristics of the target itself or an indication that the system is not specific enough to target a single-copy site in general. The fact that an integration library consisting of 21646 independent SB integrations generated by unfused SB100X without any targeting factor did not contain any integrations within 50 kb of the *HPRT* target sequence either (data not shown) might indicate that the *HPRT* gene is simply a poor target for SB integrations. It should be noted that a previous attempt to target the *piggyBac* transposase to the *HPRT* gene with CRISPR/Cas components also failed, even though targeting with other DBDs (ZFs and TALEs) was successful (*Luo et al., 2017*). Poor targeting of a single-copy chromosomal region is reminiscent of our previous findings with engineered Rep proteins (*Ammar et al., 2012*). Both Rep-SB and Rep-N57 fusions were able to enrich SB transposon integrations in the vicinity of genomic Rep binding sites, yet they failed to target integration into the *AAVS1* locus, the canonical integration site of AAV (*Ammar et al., 2012*). Thus, selection of an appropriate target site appears to be of paramount importance. The minimal requirements for such sites are accessibility by the transpositional complex and the presence of TA dinucleotides to support SB transposition; in fact, SB was reported to prefer insertion into TA-rich DNA in general (*Liu et al., 2005*). The importance of DNA composition in the vicinity of targeted sites was also highlighted in the context of targeted *piggyBac* transposition in human cells (*Kettlun et al., 2011*). Namely, biased transposition was only observed with engineered loci that contained numerous TTAA sites (the target site of *piggyBac* transposons) in the flanking regions of a DNA sequence bound by a ZF protein. An alternative, empirical approach, where careful choice of the targeted chromosomal region may increase targeting efficiencies would be to select sites where clusters of SB insertions (transposition 'hot spots') occur in the absence of a targeting factor. Targeting might be more efficient at these sites, because they are by definition receptive to SB insertions. Collectively, these considerations should assist in the design of target-selected gene insertion systems with enhanced efficiency and specificity.

The results presented here, as well as the results of previous targeting studies (*Kovač and Ivics, 2017*; *Luo et al., 2017*; *Hew et al., 2019*), indicate that the main obstacle to targeted transposition is the low ratio of targeted to non-targeted insertions. This is likely due to the fact that, in contrast to site-specific nucleases where sequence-specific DNA cleavage is dependent on heterodimerization of *Fok*I endonuclease domain monomers (*Szczepek et al., 2007*), or to Cas9, where DNA cleavage is dependent on a conformational change induced by DNA binding (*Jiang and Doudna, 2017*), the transposition reaction is not dependent on site-specific target DNA binding. The transposase component, whether as part of a fusion protein or supplied in addition to an adapter protein, is capable of catalyzing integrations without the DBD binding to its target. Thus, any attempt to target specific sites faces an overwhelming excess of non-specific competitor DNA, to which the transposase can freely bind. This non-specific binding of the transposase to human chromosomal DNA competes with specific binding to a desired target sequence, thereby limiting the probabilities of targeted transposition events. This problem might be mitigated by engineering of the transposase to reduce its unspecific DNA affinity. As SB transposase molecules have a positively charged surface (*Voigt et al., 2016*), they readily bind to DNA regardless of sequence. Decreasing the surface charge of the transposase would likely result in reduced overall activity, but at the same time it might make the transposition reaction more dependent on binding to the target DNA by the

associated DBD. The ultimate goal would be the design of transposase mutants deficient in target DNA binding but proficient in catalysis. A similar approach was previously applied to *piggyBac* transposase mutants deficient in transposon integration. Although fusion of a ZF DBD restored integration activity of the transposase in that study, enrichment of insertions near target sites specified by the DBD was not seen (*Li et al., 2013*). However, in a follow-up study, a dCas9-*piggyBac* fusion was found to enable targeted transposon integrations in the human genome, and targeting was dependent on the use of the mutant *piggyBac* transposase (*Hew et al., 2019*). Another simple modification that could potentially result in more efficient targeting is temporal control of the system. In its current form, all components of the system are supplied to the cell at the same time. It might be possible to increase targeting efficiency by supplying the targeting factor first and the transposon only at a later point to provide the targeting factors with more time to bind to their target sites.

In conclusion, this study shows that targeting SB transposon integrations towards specific sites in the human genome by an RNA-guided mechanism, though currently inefficient, is possible. This is the first time this has been demonstrated for the SB system and the first time RNA-guided transposition was demonstrated by analyzing the overall distribution of insertion sites on a genome-wide scale. If the current limitations of the system can be addressed by substantially increasing the efficiency of retargeting, and if these effects can also be observed in therapeutically relevant cell types, this technology might be attractive for a range of applications including therapeutic cell engineering. Gene targeting by HR is limited in non-dividing cells because HR is generally active in late S and G2 phases of the cell cycle (*Takata et al., 1998*). Therefore, post-mitotic cells cannot be edited in this manner (*Fung and Weinstock, 2011*; *Orthwein et al., 2015*). Newer gene editing technologies that do not rely on HR, like prime editing (*Anzalone et al., 2019*), usually have a size limitation for insertions that precludes using them to insert entire genes. In contrast, SB transposition is not limited to dividing cells (*Walisko et al., 2006*) and can transfer genes over 100 kb in size (*Rostovskaya et al., 2012*). Another drawback of methods relying on generating DSBs is the relative unpredictability of the outcome of editing. As described above, different repair pathways can result in different outcomes at the site of a DSB. Attempts to insert a genetic sequence using HR can also result in the formation of indels or even complex genomic rearrangements (*Kosicki et al., 2018*). In contrast to DSB generation followed by HR, insertion by integrating vectors including transposons occurs as a concerted transesterification reaction (*Wang et al., 2017*; *Mitra et al., 2008*), avoiding the problems associated with free DNA ends.

## Materials and methods

### Cell culture and transfection

In this work we used human HeLa (RRID:CVCL_0030), HCT116 (RRID:CVCL_0291) and HEK293T (RRID:CVCL_0063) cell lines. All cell lines originate from ATCC and have tested negative for mycoplasma. All cell lines were cultured at 37°C and 5% $CO_2$ in DMEM (Gibco) supplemented with 10% (v/v) FCS, 2 mM L-Glutamine (Sigma) and penicillin-streptomycin. For selection, media were supplemented with puromycin (InvivoGen) at 1 μg/ml or 6-thioguanine (6-TG, Sigma) at 30 mM. Transfections were performed with Lipofectamine 3000 (Invitrogen) according to manufacturer's instructions.

### Plasmid construction

All sequences of primers and other oligos are listed in *Supplementary file 1*. dCas9 fusion constructs were generated using pAC2-dual-dCas9VP48-sgExpression (Addgene, #48236) as a starting point. The VP48 activation domain was removed from this vector by digestion with *Fse*I and *Eco*RI. For dCas9-SB100X, the SB100X insert was generated by PCR amplification from a pCMV-SB100X expression plasmid with primers SBfwd_1 (which introduced the first half of the linker sequence) and SBrev_1 (which introduced the *Eco*RI site). The resulting product was PCR amplified using SBfwd_2 and SBrev_1 (SBfwd_2 completed the linker sequence and introduced the *Fse*I site). The generated PCR product was purified, digested with *Eco*RI and *Fse*I and cloned into the dCas9 vector. The dCas9-N57 construct was generated in an analogous manner, replacing primer SBrev_1 with N57rev_1 to generate a shorter insert which included a stop codon in front of the *Eco*RI site. In addition, annealing of phosphorylated oligos stop_top and stop_btm resulted in a short insert containing a stop codon and sticky ends compatible with *Fse*I- and *Eco*RI-digested DNA. Ligation of this oligo

into the *Fse*I/*Eco*RI-digested dCas9-VP48 vector resulted in a dCas9 expression plasmid. To generate the N57-dCas9 plasmid, the previously constructed dCas9 expression vector was digested with *Age*I and the N57 sequence was PCR-amplified by two PCRs (using primers SBfwd_3 and N57rev_2, followed by SBfwd_3 and N57rev_3), which introduced a linker and two terminal *Age*I sites. The *Age*I-digested PCR product was ligated into the dCas9 vector, generating a N57-dCas9 expression vector. For Cas9-SB100X and Cas9-N57 constructs, the same cloning strategy was used, using the plasmid pSpCas9(BB)−2A-GFP (Addgene, #113194) as a starting point instead of pAC2-dual-dCas9VP48-sgExpression. Insertion of sgRNA binding sequences into Cas9/dCas9-based vectors was performed by digesting the vector backbone with *Bbs*I and inserting sgRNA target oligos generated by annealing phosphorylated oligos that included overhangs compatible to the *Bbs*I-digested backbones. For expression, plasmids were transformed into *E. coli* (DH5α or TOP10, Invitrogen) using a standard heat shock protocol, selected on LB agar plates containing ampicillin and clones were cultured in LB medium with ampicillin. Plasmids were isolated using miniprep or midiprep kits (Qiagen or Zymo, respectively).

## In vitro Cas9 cleavage assay

For in vitro tests of sgRNA activities, sgRNAs were generated by PCR amplifying the sgRNA sequences with a primer introducing a T7 promoter upstream of the sgRNA and performing in vitro transcription using MEGAshortscript T7 Transcription Kit (Thermo Fisher). To test the activity of *Alu*-directed sgRNAs, 1 μg of genomic DNA isolated from human HEK293T cells was incubated with 3 μg of in vitro transcribed sgRNAs and 3 μg of purified Cas9 protein in 20 μl of 1 x NEB3 buffer (New England Biolabs) at 37°C overnight. DNA was visualized by agarose gel electrophoresis in a 1% agarose gel. After digestion, fragmented gDNA was purified using a column purification kit (Zymo) and ligated into *Sma*I-digested pUC19. The plasmids were transformed into *E. coli* DH5α and grown on LB agar supplemented with X-gal. Plasmids from white colonies were isolated and the insert ends were sequenced using primers pUC3 and pUC4. Sanger sequencing was performed by GATC Biotech. The activity of L1-directed sgRNAs was tested by digesting 100 ng of a plasmid fragment with 300 ng of purified Cas9 and 300 ng of in vitro transcribed sgRNA in 10 μl of 1 x NEB3 buffer. The DNA substrate was generated by digesting the plasmid containing a full-length L1 retrotransposon (JM101/L1.3) with *Not*I-HF (New England Biolabs) and isolating the ~3.3 kb fragment by gel extraction.

## TIDE assay

HeLa cells ($5 \times 10^6$) were transfected with the plasmid PX459/HPRT0 (co-expressing Cas9, sgHPRT-0 and a puromycin resistance cassette). After 36 hr, selection at 1 μg/ml of puromycin was applied for another 36 hr. Cells were harvested and genomic DNA was prepared using a DNeasy Blood and Tissue Kit (Qiagen). The *HPRT* locus was amplified using primers HPRT_fwd and HPRT_rev, PCR products generated from untransfected HeLa cells served as negative control. PCR products were column-purified and Sanger-sequenced using services from GATC Biotech with the primer HPRT_fwd. The sequences were analyzed using the TIDE online tool (*Brinkman et al., 2014*).

## Western blot

Protein extracts used for Western Blot were generated by transfecting $5 \times 10^6$ HeLa cells with 10 μg of expression vector DNA and lysing cells with RIPA buffer after 48 hr. Lysates were passed through a 23-gauge needle, incubated 30 min on ice, then centrifuged at 10000 g and 4°C for 10 min to remove cell debris. Total protein concentrations were determined via Bradford assay (Pierce Coomassie Plus [Bradford] Assay Kit, Thermo Fisher). Proteins were separated by discontinuous SDS-PAGE and transferred onto nitrocellulose membranes (1 hr at 100 V). Membranes were stained with α-SB antibody (RRID:AB_622119, R and D Systems, 1:500, 2 hr) and α-goat-HRP (RRID:AB_258425, Sigma, 1:10000, 1 hr) or with α-actin (RRID:AB_2223496, Thermo Scientific, 1:5000, 2 hr) and α-mouse-HRP (RRID:AB_228313, Thermo Scientific, 1:10000, 1 hr) for the loading control. Membranes were visualized using ECL Prime Western Blotting reagents.

## Transposition assay

Transposition assays were performed by transfecting $10^6$ HeLa cells with 500 ng pT2Bpuro and 10 ng pCMV-SB100X or 20 ng of dCas9-SB100X expression vector. Selection was started 48 hr post-transfection in 10 cm dishes. After two weeks, cells were fixed for two hours with 4% paraformalde-hyde, and stained overnight with methylene blue. Plates were scanned, and colony numbers were automatically determined using ImageJ/Fiji and the Colony Counter plugin (settings: size >50 px, circularity >0.7).

## Assay for Cas9 cleavage of the *HPRT* gene

For the initial validation of *HPRT*-specific sgRNAs, 1 µg each of a plasmid expressing Cas9 and separate plasmids expressing the different sgRNAs were transfected into $10^6$ HCT116 cells. For the validation of Cas9 fusion proteins, $10^6$ HCT116 cells were transfected with 3 µg plasmids expressing Cas9 (without sgRNA or with sgHPRT-0), Cas9-N57 or Cas9-SB100X (with sgHPRT-0). Selection with 30 mM 6-TG was started 72 hr after transfection. Fixing, staining and counting of colonies were performed as detailed in the previous section.

## Electrophoretic mobility shift assay (EMSA)

Nuclear extracts of HeLa cells transfected with plasmids expressing dCas9, dCas9-N57 and N57-dCas9 were generated using NE-PER Nuclear and Cytoplasmic Extraction Reagents (Thermo Fisher) according to manufacturer's instructions, and total protein concentration was determined by Bradford assay. Similar expression levels between extracts were verified by dot blot using a Cas9 antibody (RRID:AB_2610639, Thermo Fisher). A bacterial extract of N57 was used as a positive control. For the EMSA, a LightShift Chemiluminescent EMSA Kit (Thermo Fisher) was used according to manufacturer's instructions, using ca. 10 µg of total protein (nuclear extracts) or 2.5 µg of total protein (bacterial extract).

## Generation of integration libraries

SB integrations were generated by transfecting $5 \times 10^6$ HeLa cells with expression plasmids of either dCas9-SB100X (750 ng) or dCas9-N57 (9 µg) together with unfused SB100X (250 ng). All samples were also transfected with 2.5 µg of the transposon construct pTpuroDR3. For each targeting construct, plasmids containing either no sgRNA, sgHPRT-0, sgAluY-1 or sgL1-1 were used. For libraries using dCas9-N57 and dCas9-SB100X, two and six independent transfections were performed, respectively. Puromycin selection was started 48 hr after transfection and cells were cultured for two weeks. Cells were then harvested and pooled from the replicate transfections, and genomic DNA was prepared using a DNeasy Blood and Tissue Kit (Qiagen). The protocol and the oligonucleotides for the construction of the insertion libraries have previously been described (*Querques et al., 2019*). Briefly, genomic DNA was sonicated to an average length of 600 bp using a Covaris M220 ultrasonicator. Fragmented DNA was subjected to end repair, dA-tailing and linker ligation steps. Transposon-genome junctions were then amplified by nested PCRs using two primer pairs binding to the transposon ITR and the linker, respectively. The PCR products were separated on a 1.5% ultra-pure agarose gel and a size range of 200–500 bp was extracted from the gel. Some of the generated product was cloned and Sanger sequenced for library verification before high-throughput sequencing with a NextSeq (Illumina) instrument with single-end 150 bp setting.

## Sequencing and bioinformatic analysis

The raw Illumina reads were processed in the R environment (*R Development Core Team, 2017*) as follows: the transposon-specific primer sequences were searched and removed, PCR-specificity was controlled by verifying for the presence of transposon end sequences downstream of the primer. The resulting reads were subjected to adapter-, quality-, and minimum-length-trimming by the *fastp* algorithm (*Chen et al., 2018*) using the settings below: `adapter_sequence = AGATCGGAAGAGCA-CACGTCTGAACTCCAGTCAC -cut_right -cut_window_size 4 -cut_mean_quality 20 -length_required 28`. The reads were then mapped to the hg38 human genome assembly using Bowtie2 (*Langmead and Salzberg, 2012*) with the *–very-fast* parameter in *–local* mode. The 'unambiguity' of the mapped insertion site positions were controlled by filtering the sam files using SAMtools (*Li et al., 2009*) with the *samtools view –q* 10 setting. Since the mapping allowed for

mismatches the insertion sites within five nucleotide windows were reduced to the one supported by the highest number of reads. Any genomic insertion position was considered valid if supported by at least five independent reads. The genomic coordinates (UCSC hg38) of the transposon integration-site sets of all the conditions are provided as *Figure 5—source datas 1*, *2*, *3* and *4*. Insertion site logos were calculated and plotted with the SeqLogo package. The frequencies of insertions around the sgRNA target sequences were displayed by the genomation package (*Akalin et al., 2015*). Probability values for nucleosome occupancy in the vicinity of *Alu*Y targets and non-targeted insertion sites were calculated with a previously published algorithm (*Segal et al., 2006*).

## Statistical analysis

Significance of numerical differences in transposition assay and Cas9 cleavage assays was calculated by performing a two-tailed Student's t-test using the GraphPad QuickCalcs online tool. All experiments that have colony numbers as a readout were performed in triplicates. We used the Fishers' exact test for the statistical analyses of the TA-target contents and the frequencies of insertion sites in various genomic intervals.

## Acknowledgements

We thank T Diem for technical support.

## Additional information

### Competing interests

Zoltán Ivics: Patent applications around targeted transposon integration technology (Patent Nos. EP1594971B1, EP1594972B1 and EP1594973B1). The other authors declare that no competing interests exist.

### Funding

No external funding was received for this work.

### Author contributions

Adrian Kovač, Data curation, Formal analysis, Validation, Investigation, Visualization, Methodology, Writing - original draft; Csaba Miskey, Software, Investigation, Visualization; Michael Menzel, Software, Investigation; Esther Grueso, Investigation, Methodology; Andreas Gogol-Döring, Software, Supervision, Investigation; Zoltán Ivics, Conceptualization, Resources, Data curation, Supervision, Funding acquisition, Project administration, Writing - review and editing

### Author ORCIDs

Adrian Kovač ⓘ https://orcid.org/0000-0001-7644-8918
Csaba Miskey ⓘ https://orcid.org/0000-0002-7566-9556
Michael Menzel ⓘ https://orcid.org/0000-0002-4129-4741
Zoltán Ivics ⓘ https://orcid.org/0000-0002-7803-6658

### Decision letter and Author response

Decision letter https://doi.org/10.7554/eLife.53868.sa1
Author response https://doi.org/10.7554/eLife.53868.sa2

## Additional files

### Supplementary files

- Supplementary file 1. Sequences of DNA oligos used in this study.
- Transparent reporting form

## Data availability

DNA sequence data generated and analysed during this study are included in the manuscript and Figure 5—source data 1–4.

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
