## [Decision Letter]

**Acceptance summary:**

This is a very interesting beginning for the ability to target transposition. I was very impressed with the careful approach and the additional work you provided in response to reviewers. I am very pleased to see this working in mammalian cells and look forward to future improvements.

**Decision letter after peer review:**

Thank you for submitting your article "RNA-guided Retargeting of *Sleeping Beauty* Transposition in Human Cells" for consideration by *eLife*. Your article has been reviewed by three peer reviewers and the evaluation has been overseen by Didier Stainier as the Senior Editor. The reviewers have opted to remain anonymous.

The reviewers have discussed the reviews with one another and the Reviewing Editor has drafted this decision to help you prepare a revised submission.

Summary:

Delivery and safe insertion of therapeutic genes is challenging. The manuscript by Kovac et al. outlines a strategy to target *Sleeping Beauty* (SB) transposon integration to specific regions of the genome by fusing a dead Cas9 (dCas9) to the SB transposase. SB integrates somewhat randomly through the genome and has been used for gene therapy applications. The strategy, if successful, would allow clinicians to increase the safety of SB gene therapy through targeting transposition to a safe harbor(s) in the genome without creating a double strand break. Such a strategy would likely have a large impact. There was general concern over the minimal effects of targeting and whether the observed targeting was relevant biologically.

The authors nicely examine the activity using different strategies for SB targeting and examine different chimeric proteins for activity. The authors targeted *HPRT* as a single locus and *Alu* repeats as multiple copy approach. Elegant bioinformatic analyses are used to examine integration sites across the genome. Whilst the targeting to a single gene, *HPRT* was not successful, the targeting to an *Alu* repeat did show some significant enrichment of insertions in close proximity of the targeted sequence. Given that the chromatin surrounding the sgRNA targeted insertion site is naturally disfavoured by SB transposase, this suggests that targeting might be underestimated. Although the technology clearly requires further optimization, this is a proof-of-concept study with very exciting and promising results and an important step towards developing safer gene therapy vectors. The majority of transposon integrations still occur randomly. This activity was highest 200 bp away from the targeting site. This represents a significant step forward, but as the authors note in the Discussion, the overall strategy could be difficult to improve. Transposition is not controlled upon binding the targeted site. The impact is tempered by the low level of targeting observed.

Essential revisions:

The experiments must be repeated with using the sgRNA *HPRT* as control in addition to the transposition reaction without the sgRNA. If the same effects are observed, the claims through the paper should still be downplayed due to the low level of targeting.

The authors claim in the Abstract and the Discussion to demonstrate biased integration by combining SB with dCas9 for All, however this is not supported by the data in my opinion, and also clearly indicated by the authors (subsection “RNA-guided *Sleeping Beauty* transposition in the human genome”, third paragraph). However, at several occasions the authors claim to show proof-of-principle that the pattern of transposition can be influenced.

Throughout the paper the conclusion is 'over' claimed:

- in the Abstract the claims should be downplayed substantially;

- last sentence of Abstract should be omitted;

- last sentence of the Introduction should be downplayed;

- in the Discussion (first and last paragraphs for example).

In the paper (especially in the Introduction) the word viruses is used, where the author refer to viral vectors (or virus-derived vectors). For clarity, please replace (in Introduction, second and third paragraphs, etc.).

It is not clear whether independent transposition experiments were done, and if so, how many. Please indicate more clearly how many were done, and how many unique integrations were retrieved for each experiment.

In the subsection “Design and validation of sgRNAs targeting single-copy and repetitive sites in the human genome” the authors demonstrate functionality of the sgRNAs used. In order to show efficiency of the authors could specifically amplify the CRISPRsgRNA targeted region, sequencing and subsequent TIDE analysis. The current assays are indicative, but do not 'establish' these sgRNA as efficient.

In Figure 4C: how do the authors calculate a SEM for n=2 experiment?

Also for Figure 4C: sgRNA0 also shows an efficiency of 60 6-TG resistant clones, whereas earlier 300 clones were observed (Figure 2C). Why this variation?

sgHPRT data are not shown (subsection “RNA-guided *Sleeping Beauty* transposition in the human genome”). Please indicate as such or leave out. What is the control, is this without sgRNA?

For the retargeting experiment (Figure 5) control sgRNAs would be interesting to use. Alternatively, the authors could use the integration data obtained for the HPRT sgRNA as a control.

For Figure 5: it is not clear from the setup of the tables in Figure 5B and C what the /*Alu*Y means. Indicate more clearly that this refers to sgRNA-*Alu*Y. Please include the sgRNA *HPRT* data here as an unrelated control. Also, why is there a different plot for the blue line in Figure 5E and F (even though this is twice the same experiment)?

Plotting integration frequencies (Figure 5D) shows that the effects that can be obtained are marginal, and this should be stated as such. How many independent experiments were done? Is the effect observed reproducible?

The authors only show data for HeLa cells; the claim that the technology may be used for therapeutic cell engineering is wrong (Discussion, last paragraph) and should be deleted.

---

## [Author Response]

Essential revisions:The experiments must be repeated with using the sgRNA HPRT as control in addition to the transposition reaction without the sgRNA. If the same effects are observed, the claims through the paper should still be downplayed due to the low level of targeting.

In order to address the question whether targeting to *Alu*Y can also be observed in comparison to an unrelated sgRNA, we replaced the –sgRNA control with another control dataset that we obtained with an sgRNA targeting another repetitive element, the L1 retrotransposon. We believe that the dataset generated by sgL1 is superior to a “no sgRNA” (-sgRNA) control and to a HPRT (sgHPRT) control, because sgL1 targets a multi-copy sequence (like sgAluY), and because the total numbers of insertions obtained with this sgRNA were comparable to those obtained with sgAluY (the library generated with dCas9-SB100X + sgHPRT-0 had very few overall insertion sites). We therefore opted to use this control dataset, and present new analysis in our revised manuscript.

Regarding the claims we make throughout the paper, we do not believe that they are ‘overclaimed’. We acknowledge, throughout the text, that the extent, to which sgAluY in conjunction with dCas0-SB100X biases target site selection of SB is small. We do not claim any direct practical application of this system in its current form. Nonetheless, the address the concerns of the reviewer directly, some of the claims that reviewers perceived to be too strong were slightly modified. Please see points below for further clarification.

The authors claim in the Abstract and the Discussion to demonstrate biased integration by combining SB with dCas9 for All, however this is not supported by the data in my opinion, and also clearly indicated by the authors (subsection “RNA-guided Sleeping Beauty transposition in the human genome”, third paragraph). However, at several occasions the authors claim to show proof-of-principle that the pattern of transposition can be influenced.

We believe that the data support that the integration pattern can be influenced by constructs containing RNA-guided DBDs, even though the effects are currently very small. The claim in the Discussion specifically refers to targeting with the dCas9-N57 adapter protein. See also responses below.

Throughout the paper the conclusion is 'over' claimed:- in the Abstract the claims should be downplayed substantially;

This passage reads like this: “Here we demonstrate biased genome-wide integration of the *Sleeping Beauty* (SB) transposon by combining it with components of the CRISPR/Cas9 system. We provide proof-of-concept that it is possible to influence the target site selection of SB by fusing it to a catalytically inactive Cas9 (dCas9) and by providing a single guide RNA (sgRNA) against the human *Alu* retrotransposon.”We believe this text summarizes some of our conclusions in a factual matter. We state that we used components of CRISPR/Cas (which is a fact, not an opinion). We believe the word “bias” describes in this context a difference in the insertional patterns of SB transposons in the presence of different sgRNAs. The use of the word “bias” is not dependent, in our opinion, on whether such difference is large or small. Finally, we factually state that this difference was seen by using a fusion protein consisting of dCas9 and the SB100X transposase and an sgRNA that targets the *Alu* retrotransposon. We also use the word “influence”. Again, we detect a bias (or difference) and we conclude that this bias (or difference) is dependent on the fusion protein and on the sgRNA against *Alu*. If the bias (or difference) is dependent on these molecules, then we can conclude that these factors have an influence.

- last sentence of Abstract should be omitted;

This sentence reads like this: “Future modifications of this technology may allow the development of methods for specific gene insertion for precision genetic engineering.” We believe this is a fairly neutral statement, and simply indicates to the reader the context, for which the work has relevance. We are certainly not claiming that we have reached this goal; instead, we clearly acknowledge that modifications are required for this technology to become useful.

- last sentence of the Introduction should be downplayed;

The last sentence of the Introduction currently reads like this: “Here, we present proof-of-principle evidence that integrations of the SB transposon system can be biased towards endogenous *Alu* retrotransposons using dCas9 as a targeting domain in an sgRNA-dependent manner.” This sentence delivers a summary statement that formulates a take-home message, by slightly different (but nonetheless scientifically justifiable) wording than in the Abstract.

- in the Discussion (first and last paragraphs for example).

We have included some modifiers in the text to explicitly state that the effects that we see are small, and that future modifications of the system will be required for utility.

In the paper (especially in the Introduction) the word viruses is used, where the author refer to viral vectors (or virus-derived vectors). For clarity, please replace (in Introduction, second and third paragraphs, etc.).

We thank the reviewer for this observation. The first instance was not replaced, as this sentence explains that the relevant vectors were based on these two types of integrating viruses. The word ‘from’ was added to the sentence to clarify this point. In the other two instances, the appropriate replacements were made.

It is not clear whether independent transposition experiments were done, and if so, how many. Please indicate more clearly how many were done, and how many unique integrations were retrieved for each experiment.

The numbers of independent transposition experiments (independent transfections) are now explicitly stated in the revised manuscript in Materials and methods section. The numbers of unique integrations for each experiment is indicated in Figure 5B, and stated in the subsection “RNA-guided *Sleeping Beauty* transposition in the human genome”.

In the subsection “Design and validation of sgRNAs targeting single-copy and repetitive sites in the human genome” the authors demonstrate functionality of the sgRNAs used. In order to show efficiency of the authors could specifically amplify the CRISPRsgRNA targeted region, sequencing and subsequent TIDE analysis. The current assays are indicative, but do not 'establish' these sgRNA as efficient.

Many thanks for this great suggestion. A TIDE assay was performed to assess the efficiency of sgHPRT-0. We chose to concentrate on HPRT for this analysis, because sgAluY-1 has a high number of binding sites in the genome and binds very close to the end of the *Alu*Y sequence (ca. 20 bp away from the 5’-end of *Alu*). Thus, it is technically impossible to generate a PCR product of the required size for the TIDE assay.

In Figure 4C: how do the authors calculate a SEM for n=2 experiment?

The experiments described in Figure 2C and Figure 4C were repeated in triplicates in order to improve the statistical analysis.

Also for Figure 4C: sgRNA0 also shows an efficiency of 60 6-TG resistant clones, whereas earlier 300 clones were observed (Figure 2C). Why this variation?

The variation in the previous experiments was likely due to different transfection efficiencies, as different transfection reagents were used. We have repeated these experiments using the same transfection conditions for our revised submission, resulting similar numbers of drug-resistance cell colonies. However, it should be noted that the Cas9 + sgHPRT-0 samples in these two experiments have not been done under the exact same condition: in Figure 2, the sgRNA is expressed from a separate plasmid, while in Figure 4 the sgRNA is expressed from the same plasmid as Cas9.

sgHPRT data are not shown (subsection “RNA-guided Sleeping Beauty transposition in the human genome”). Please indicate as such or leave out. What is the control, is this without sgRNA?

sgHPRT-0 data not being shown is now indicated and the relevant paragraph was significantly shortened.

For the retargeting experiment (Figure 5) control sgRNAs would be interesting to use. Alternatively, the authors could use the integration data obtained for the HPRT sgRNA as a control.

Please see response to comment #1.

For Figure 5: it is not clear from the setup of the tables in Figure 5B and C what the /AluY means. Indicate more clearly that this refers to sgRNA-AluY. Please include the sgRNA HPRT data here as an unrelated control. Also, why is there a different plot for the blue line in Figure 5E and F (even though this is twice the same experiment)?

The combinations of sgRNAs and targeting constructs are now clearly indicated in the table. Regarding the use of an unrelated control, please see our response to comment #1. The use of different window sizes in Figure 5E and F (Figure 5D and E in the revised figure) was necessary to avoid empty windows in the control datasets when applied to mismatched targets. This would have required a division by zero, thus, window sizes were increased to ascertain that all control windows contained at least one insertion.

Plotting integration frequencies (Figure 5D) shows that the effects that can be obtained are marginal, and this should be stated as such. How many independent experiments were done? Is the effect observed reproducible?

The modest, and statistically insignificant effect shown in Figure 5D (Figure 5C in the revised figure) is now explicitly stated in the Results section (subsection “RNA-guided *Sleeping Beauty* transposition in the human genome”) and in the figure legend. The integration sites making up these libraries are generated from cells obtained from several independent transfections (see also response to comment 5). We have stated this explicitly in the revised manuscript in Materials and methods. Thus, our insertion libraries represent pools obtained from several independent transfections.

The authors only show data for HeLa cells; the claim that the technology may be used for therapeutic cell engineering is wrong (Discussion, last paragraph) and should be deleted.

A qualifying statement was added to the last paragraph of the Discussion to clarify that potential utility is not claimed for the system in its current form. We make a clear statement that (i) efficiency of targeting would need to be substantially increased and (ii) targeting would need to be demonstrated in therapeutically relevant cell types for future utility in therapeutic cell engineering.